# Ultrasonic Microbubble Cavitation Enhanced Tissue Permeability and Drug Diffusion in Solid Tumor Therapy

**DOI:** 10.3390/pharmaceutics14081642

**Published:** 2022-08-06

**Authors:** Jide He, Zenan Liu, Xuehua Zhu, Haizhui Xia, Huile Gao, Jian Lu

**Affiliations:** 1Department of Urology, Peking University Third Hospital, Beijing 100191, China; 2Key Laboratory of Drug-Targeting and Drug Delivery System of the Education Ministry, Sichuan Engineering Laboratory for Plant-Sourced Drug and Sichuan Research Center for Drug Precision Industrial Technology, West China School of Pharmacy, Sichuan University, Chengdu 610064, China

**Keywords:** ultrasound, microbubbles, cavitation effect, permeability, EPR effect, tumor therapy

## Abstract

Chemotherapy has an essential role not only in advanced solid tumor therapy intervention but also in society’s health at large. Chemoresistance, however, seriously restricts the efficiency and sensitivity of chemotherapeutic agents, representing a significant threat to patients’ quality of life and life expectancy. How to reverse chemoresistance, improve efficacy sensitization response, and reduce adverse side effects need to be tackled urgently. Recently, studies on the effect of ultrasonic microbubble cavitation on enhanced tissue permeability and retention (EPR) have attracted the attention of researchers. Compared with the traditional targeted drug delivery regimen, the microbubble cavitation effect, which can be used to enhance the EPR effect, has the advantages of less trauma, low cost, and good sensitization effect, and has significant application prospects. This article reviews the research progress of ultrasound-mediated microbubble cavitation in the treatment of solid tumors and discusses its mechanism of action to provide new ideas for better treatment strategies.

## 1. Introduction

Despite the development and research of anti-tumor therapy having made obvious progress, we are compelled to attach great efforts to the drug resistance in patients undergoing systematic chemotherapy [1]. The occurrence of chemotherapy resistance has been directly related to the interaction between genetic factors, cellular heterogeneity, and tumor microenvironment (TME) [2,3,4]. Evidence demonstrated that the overall 5-year survival rate of cancer patients has increased from 49% in 1970 to the current 68% and the overall mortality rate of patients has decreased by 32% aroused by the continuous breakthroughs in medical technology and targeted drugs [5]. Disconcertingly, the gradual improvement in the survival period and mortality of cancer patients promotes the continuous prolongation of the anti-tumor treatment courses, which greatly aggravates the increase in anti-tumor drug resistance and toxic side effects with a serious threat to life quality and prognosis [6,7,8]. Therefore, how to reverse drug resistance, improve side effects, and promote chemotherapy sensitization effects are the key issues to solve urgently.

Most of the anti-tumor drugs and delivery vehicles used in current research do not have tumor selectivity themselves. It is difficult to obtain a satisfactory drug accumulation effect at tumor sites only by inherent EPR effect through passive targeted transport that ultimately leads to inefficient drug efficiency and serious side effects in clinical trials [9]. In recent years, researchers have proposed a variety of methods to facilitate the targeted delivery of drugs to tumor sites. Among them, the ultrasound-mediated microbubble cavitation effect has attracted the attention of many researchers. Using ultrasound to mediate the cavitation effect of microbubbles can enhance tissue and vascular permeability and targeted drug retention, which is an effective way to improve the EPR effect. Compared with the traditional targeted drug delivery scheme, the microbubble cavitation effect has huge advantages due to its smaller trauma, lower cost, and better sensitization effect [10] with significant application prospects. This review summarizes the progress of the ultrasound-mediated microbubble cavitation effect in solid tumor therapy by effectively improving drug delivery obstacles via the EPR effect in TME, discusses the potential mechanisms of microbubble cavitation enhancing the EPR effect, and ultimately describes the new ideas based on the therapeutic research in vitro and in vivo for better therapeutic strategies.

## 2. EPR Effect in the TME

### 2.1. Chemotherapy Obstacles Aroused by Aberrant Angiogenesis in TME

Although many chemotherapy agents have ideal effectiveness on tumor cells cultured in vitro, the unique aberrant angiogenesis in TME of tumors in vivo has multiple obstacles that restrict drug efficacy; thus, the provision of traditional chemotherapy is suboptimal [11,12,13,14,15]. The solid TME consists of tumor cells, stromal cells, and secreted cytokines. Compared with normal tissue cells under stable conditions maintained by numerous regulatory factors, tumor cells have characteristics of rapid proliferation, cell cycle disorders, and immune escape [16,17]. Vascular development in TME is necessary for the survival and progression of tumor cells and has an important impact on the remodeling of the TME, and provides a conducive environment for tumor growth and invasion [18,19]. With the rapid proliferation of tumor cells, various angiogenesis regulators are secreted in TME to induce the generation of a neovascular network [18,20]. Vascular development mainly includes three processes: angiogenesis, neovascularization, and vascular remodeling [21,22,23]. Due to the overexpression of pro-angiogenic factors induced by rapid tumor proliferation, tumor neovascularization with abnormal structure and function could not achieve normal oxygen supply of tissue cells leading to a TME surrounded by hypoxia and ischemia [24,25,26]. Remolded TME will further aggravate angiogenesis abnormality and increase intra-tumoral fluid pressure causing reduced blood perfusion supply, eventually inhibiting immune cells from exerting anti-tumor efficiency and impeding chemotherapeutic drug diffusion [25,27,28,29,30]. Meanwhile, imperfect vascularization causing disorder distribution and irregular anastomotic branches combined with discontinuous vascular structure connections in the tumor could induce the lost blood–tumor barrier function, weaken vessel stability, and impair blood perfusion regulation [31,32,33]. In addition, the discontinuous structure between the basement membrane with pericytes and endothelial cells results in leakage and inefficiency in blood perfusion [27,29]. In the central region of the tumor, intra-tumoral high interstitial fluid pressure (IFP) can lead to vascular perfusion decrease, hypoxic state, and lymphatic vessel compression [14,34,35,36]. Thus, immature tumor vessels in TME have characteristics of vascular leakage, decreased blood perfusion, structure connection impairment, barrier dysfunction, and drug diffusion diminishment. Additionally, the increased vascular leakage without effective lymphatic drainage inside the tumor eventually leads to a significant increase in IFP in the tumor, which dramatically hinders the diffusion of therapeutic drugs in the interstitial space [37,38,39,40,41]. In addition, with the rapid proliferation of tumoral cells, the increased dispersal distance between tumoral cells and neovessels further hinders the delivery of chemotherapeutic drugs from blood vessels into interstitium, leading to exacerbating the formation of chemotherapy resistance [42,43,44].

### 2.2. Therapeutic Strategy Based on the EPR Effect

The TME in solid tumors presents several disorders with vascular tortuosity, angiogenesis, and hypoxia [45,46,47]. Meanwhile, abnormal tissue vasculature, stromal matrix, and interstitial fluid pressure are major limitations preventing drug penetration in solid tumors [34,48]. Therefore, therapeutic agents transported into tumor lesions in the accurate and efficient delivery is crucial for effectively addressing drug delivery in TME which remains a major challenge to date. Fortunately, the lack of vascular supporting tissue in tumors results in leaky vessels and poor lymphatic drainage is a structural basis for enhancing the effect of EPR [49]. The EPR effect in preclinical research with passive targeted delivery systems has shown significant enhancement in anticancer efficacy compared with traditional chemotherapeutics [50,51,52]. However, due to tumor heterogeneity with aberrant vascular permeability, phenotypic mutation, and stromal abnormalities hinder the successful application of EPR-based therapy [53,54,55,56,57]. Therefore, precision therapy and complementary enhancement of EPR to overcome chemotherapy resistance challenges aroused by aberrant angiogenesis in TME are crucial for the development of personalized EPR-mediated solid tumor therapy [58].

## 3. Ultrasonic Microbubble Cavitation Enhancing the EPR Effect on Tumor Therapy

### 3.1. Mechanical Effects of Ultrasonic Microbubble Cavitation

The ultrasound-mediated microbubble cavitation effect refers to a series of dynamic processes such as expansion, contraction, oscillation, and violent collapse in tiny bubbles under ultrasound sonication at a specific frequency [59,60,61]. The physical phenomena with high temperature, high pressure, and micro-jet are induced by rapid release of instantaneous energy accompanying ultrasound intervention in bubbles after energy absorption [62,63]. The acoustic cavitation effect is commonly used to improve cancer therapy for cancers such as hepatocellular carcinoma, colon cancer, brain cancers, and so on, in clinical research [64,65,66]. When ultrasound is applied to the current and emerging technique on diagnosis and treatment, the cavitation effect can improve EPR in terms of both drug release and biological effects. During the process of agent release, ultrasound can stimulate the carrier to enhance the efficiency of drug release and distribution in tissues. In terms of biological effects, the use of acoustic energy combined with the cavitation effect of microbubbles is mainly to promote the vascular permeability and extracellular material transport by membranes in TME [67,68,69].

When exposed to an ultrasound field, the liquid around the microbubbles will form acoustic streaming, in which the shear stress generated by the streaming can promote the directional release of the drug from microbubbles and enhance the permeability of the drug in tissues and cell membranes (shear-induced permeability) [68]. These mechanical effects primarily generated by ultrasound may lead to the improvement in permeability and vascular perfusion into solid tumor tissues [70].

### 3.2. Ultrasonic Microbubble Cavitation Promoting Tumor Therapy by Enhancing the EPR Effect

Microbubbles can present distinct oscillation patterns when acoustic parameters are varied. The stable ultrasonic cavitation effect of microbubbles is usually generated under relatively low peak negative acoustic pressures. When the acoustic pressure amplitudes were further increased, ultrasonic cavitation turned into a violent collapse called inertial cavitation accompanied by shock waves and microstreaming [71]. These effects of cavitation have been extensively applied in various domains of medical applications [72,73,74,75].

Ultrasound-mediated microbubble cavitation can enhance EPR by improving the permeability of the biological barriers in TME through a local controllable cavitation effect, enhance material exchange and transport, and achieve the therapeutic effect of increasing drug concentration in tissue cells (Figure 1). The hydrophobic microbubbles encapsulated by lipid, protein, or polymer shell gas are irradiated by ultrasound to expand, contract, oscillate, and even violently collapse, a process called cavitation. Cavitation can be divided into two forms, namely steady-state cavitation and inertial cavitation. Steady-state cavitation refers to the stable vibration of microbubbles around the resonant diameter at low sound intensities. As the sound intensity increases, the microbubbles expand, contract, or collapse more violently, generating shock waves and microjets near the microbubbles, a process called inertial cavitation [76]. In recent decades, numerous studies have demonstrated that ultrasound-mediated cavitation of microbubbles can facilitate the delivery of anticancer drugs to tumor cells [77,78]. Microbubbles are composed of less than 100 nm layers of polymers, proteins, and lipids covering the surface of a hydrophobic fluorinated gas. To increase tumor tissue specificity, ligands for specific cell surface receptors can be attached to the micro-vesicles. Due to the sound pressure of ultrasonic waves, the microbubbles will shrink and expand periodically. When the sound pressure reaches a certain threshold, the microbubble collapses [79]. The cavitation or explosion of microbubbles will form temporary holes in the cell membrane and blood vessel wall to enhance permeability, providing temporary and reversible channels for the transport of substances, so that therapy agents can enter the cell passively [80,81]. Studies have shown that ultrasound could not only enhance the passive diffusion of drug, but also affect the active transport for enhancing drug uptake. Ultrasound cavitation is thought to induce changes in ion channels and molecular reaction including membrane resealing or gap restoration in different spatial and temporal scales, resulting in increased intracellular Ca^2+^ concentration and cytoskeletal rearrangement [82]. These changes discussed above could play a crucial role in stimulating the clathrin-mediated endocytosis pathway to promote drug diffusion into cells [83]. In addition, fluid movement caused by cavitation may also facilitate vascular perfusion according to vasodilator expression including nitric oxide induced by increased intracellular Ca^2+^ concentration and high shear stress from oscillating bubbles, increasing the amount of drug agent uptake by distant tumor cells [83,84,85]. 

In recent years, many studies have also confirmed that claudins and ZO-1 play an important role in the permeability regulation of biological barriers such as the blood–tumor barrier [86,87,88,89,90]. Studies have shown that there are a series of intercellular junctions between endothelial or epithelial cells, of which tight junctions are the most important. Between mammalian cells, tight junctions are mainly composed of transmembrane proteins (Occludin), claudins, junctional adhesive molecules (JAMs), and cytoplasmic attachment proteins (ZO family) [91,92,93]. Tight junctions widely exist in biological barrier structures such as intestinal mucosal epithelial cells, interstitial cells, the blood–testis barrier, and the blood–brain barrier [94,95,96,97]. Through research, Bae et al. found that the permeability of tight junctions in the barrier structure increased after physical treatment of the biological barriers, and the expression and distribution of the main components of tight junctions changed [98]. Through research, it was found that after intravenous injection of contrast agent microbubbles, low-frequency ultrasound irradiation can significantly increase the drug concentration in the tissue, and 24 h after the ultrasound irradiation, the drug concentration was significantly reduced, and tissue cells were observed by transmission electron microscopy. The gap between them widened and recovered after 24 h. Changes in the spatial structure of tight junctions are temporally consistent with changes in tissue barrier permeability. This indicates that tight junctions play an important role in the regulation of tissue cell permeability. Studies have shown that the tight junction protein Occludin plays an important role in the process of low-frequency ultrasound irradiation combined with contrast agent microbubbles to open the blood–tumor barrier (Figure 2). When the blood–tumor barrier is opened, the expression level of this tight junction protein decreases [99]. This indicates that the tight junction protein Occludin plays an important role in enhancing the permeability of tissue cells by low-frequency ultrasound irradiation combined with contrast agent microbubbles.

Overall, ultrasound can utilize the local microbubble cavitation to enhance the EPR effect for non-invasive targeted therapy of diseases without affecting the surrounding soft tissues. The cavitation effect can achieve the passive agents’ diffusion in tissue through ameliorating the permeability of tissue and vascular barrier by sonoporation and regulate the intercellular tight junction. Similarly, it can also enhance intracellular uptake via stimulating the clathrin-mediated endocytosis pathway induced by ion channels. On the one hand, the cavitation effect can also enhance the drug delivery efficiency by inducing the releasing vasodilator to a certain extent to increase local tissue blood perfusion. The feasibility and potential of this approach might contribute to achieve better targeted delivery in the prospective fast-revolutionizing disease area.

### 3.3. Application Studies Using Ultrasonic Microbubble Cavitation on Tumor Therapy

Many studies have confirmed that ultrasound-mediated microbubble cavitation can promote drug diffusion and induce tumor-suppressive effects by enhancing EPR through improving permeability and vascular perfusion in vitro (Table 1) and in vivo (Table 2), manifested as tumor growth inhibition, increased tumor cell apoptosis and necrosis, decreased tumor angiogenesis, and decreased expression of tumor-associated proteins. Nevertheless, a profusion of adverse effects has also been reported including hemorrhage, thrombus formation, local burns, tissue necrosis, and various organ toxicities [100,101,102]. Thankfully, most of the serious side effects of ultrasound-assisted therapy including necrosis and hemorrhage, are concentrated in a relatively high intensity focused ultrasound which mainly exerts an acoustic thermal effect rather than cavitation effect. Thus, keeping the ultrasonic waves under a lower intensity level and shorter intervention time could induce the controllable cavitation effect without obvious cell death and thermal damage [103]. Overall, low-intensity ultrasound is a relatively safe non-invasive intervention strategy. It is worthwhile to expect that flexible regulation of ultrasonic intervention parameters and better protocol design can further improve cavitation effect efficacy and potential risk aversion [104].

Meanwhile, complex models are crucial to represent the in vivo TME better which can provide a unique opportunity to study cellular interactions and biophysical mechanisms involved which are difficult to replicate in vitro due to lacking intricate intracellular and intercellular signaling pathways. It is therefore important to use more experimental efforts to comprehend the inherent differences between in vitro and in vivo that will affect microbubble behavior for exploring effective treatment interventions.

## 4. Challenges and Perspectives of Ultrasonic Microbubble Cavitation in Tumor Therapy

The TME is a complex and variable milieu in which it is a key point of efficacious therapy to overcome obstacles existing in solid tumors for successfully targeting therapeutic agents to the tumor site. In recent years, a variety of tumor-targeted drug delivery systems, including nanogels, liposomes, microbubbles, magnetic nanomaterials, etc., have been researched and developed to enhance tissue permeability and facilitate drug diffusion for antineoplastic therapy. This review focuses on the therapeutic role of ultrasound-mediated microbubble cavitation by enhancing TME permeability and promoting drug diffusion in solid tumors. Although these applications in this sector have realized significant progress, there are still many difficulties and challenges which require further efforts to explore more suitable delivery systems and effective efficacy.

Firstly, the particle size of microbubbles remains a key factor affecting localized drug accumulation and cavitation effects in tumors. Currently, many researchers choose to encapsulate the therapeutic agents inside the microbubble for controllable and targeted release. Studies have shown that liposomes with particle sizes from 100 to 200 nm can accumulate agent concentration four times greater than particle sizes smaller than 50 nm or larger than 300 nm in tumors [118]. Therefore, we need to develop a new process to solve the situation that the cavitation effect is weakened due to the low accumulation of microbubbles around the tumor tissue caused by the unsuitable particle size of microbubbles.

Secondly, with the emergence of multifunctional drug delivery systems, the structure with membrane shells continues to present complex trends. We need to reduce adverse effects of membrane shell material, particle size, and targeted modification type on the cavitation effect in the complex and variable TME, and optimize the best-suited parameters of different drug-loading systems in realizing the cavitation effect for achieving the controllability and stability of the targeted microbubble-loading system between different individuals with finally attaining the standardized application in clinical intervention.

Of course, due to the different research directions of each scholar, there are also differences in the parameters they used. The optimization of the parameters used for the ultrasound-mediated microbubble cavitation effect makes it very important to apply this technology to the clinic. Therefore, we need more efforts to verify the effects of ultrasound intervention time, irradiation time interval, ultrasound frequency, sound intensity, microbubble size, drug dose, and concentration on the therapeutic efficacy of the disease.

Meanwhile, the integration of multiple technologies needs to be carried out, including protein-membrane-targeted modification, photothermal, magnetic field, radiation, free radicals, gene interference, immunotherapy, etc., to comprehensively enhance the anti-tumor efficacy.

Absolutely, the safety of antitumor drugs is also an issue that closer attention should be paid to. Ultrasound-mediated microbubble cavitation accompanied by tissue microvascular damage and thrombosis could cause systematic tissue adverse effects, similarly, the toxicity of intermediate metabolites and degradation products of the microbubble-loading system should also attract our attention. Although many studies have confirmed the high histocompatibility of microbubbles, more research is still needed in the future to further confirm the potential harm caused by long-term accumulation in the body.

At present, there are still many problems to be solved in the treatment of tumors with low-frequency ultrasound combined with microbubbles, but it is undeniable that this technology has shown great clinical application value as a safe, effective, easy-to-operate, and targeted non-invasive treatment method. With the development of technology, this promising non-invasive tumor treatment method will be widely used in clinical practice.

## 5. Conclusions

The development of tumor therapy relies heavily on the development of non-invasive drug delivery methods to efficiently and selectively deliver drugs to target cells while minimizing the systemic toxicity of drugs, which is especially the focus of future research. Microbubble-mediated ultrasonic cavitation has unique advantages in the study of enhanced drug delivery to improve the efficiency of tumor treatment. The application of ultrasound has certain advantages, such as good tolerance, non-toxicity, easy operation and implementation, and low dependence on instruments and equipment. Many studies have confirmed that thermal and cavitation effects play a major role in various treatments; however, researchers should be aware that the anti-tumor effects of various ultrasound-mediated treatments are produced by a combination of multiple biological effects. Meanwhile, the safety issues generated by ultrasonic cavitation with EPR effects enhancement including toxicity and tissue damage still need to be fully investigated as a priority for future clinical applications.

Although both diagnostic ultrasound and contrast agents are generally recognized as safe and approved for widespread use in clinical diagnosis, the combination of ultrasound and microbubble therapy has some limitations. Recently, several preclinical studies have used small animals to evaluate the safety of ultrasound combined with microbubbles using parameters such as body weight, dietary habits, and mobility [119,120]. However, the specific mechanism of the synergistic effect of ultrasonic cavitation combined with chemotherapeutic drugs is still unclear, the current sample size of relevant clinical studies is small, and the efficacy and safety of combined therapy need more studies with a larger sample size for support. Before the new technology of microbubble-enhanced ultrasonic cavitation combined with drug therapy for tumors is widely used in the clinic, more systematic and comprehensive side effects studies on various treatment options are still needed.

With the further development of molecular targeting technology, the preparation of more new microbubbles, the exploration of new treatment modes, the development of new ultrasonic equipment, and the optimization of new treatment procedures, will provide an opportunity for the treatment of tumors by microbubble-enhanced cavitation combined with drugs. This new technology brings more possibilities and brings more benefits to clinical oncology treatment programs.

## Figures and Tables

**Figure 1 pharmaceutics-14-01642-f001:**
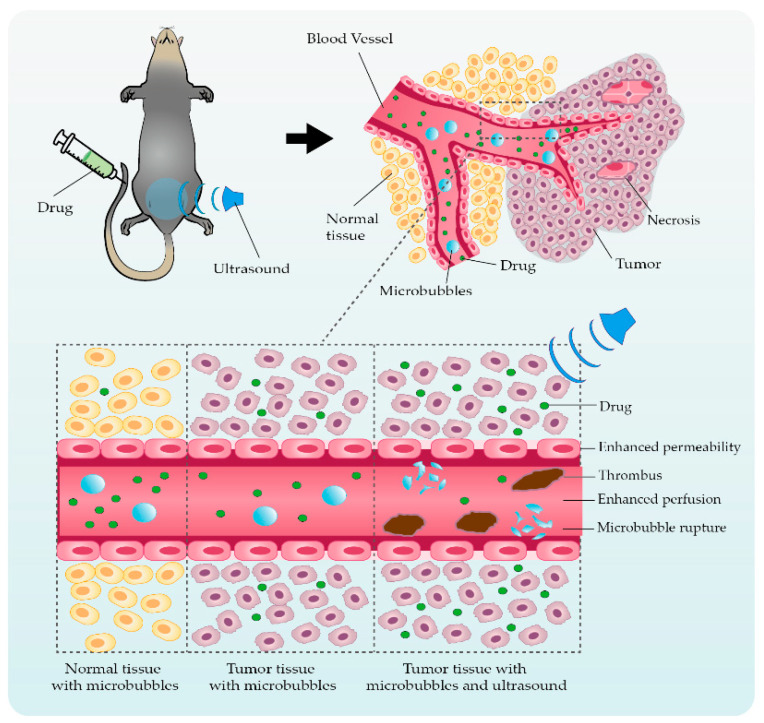
Ultrasound-mediated microbubble cavitation enhances biological barriers’ permeability and material transport. High interstitial pressure aroused by lack of blood perfusion and lymphatic return in solid tumors hinders the uptake and absorption of drug agents in cells. Using ultrasound to mediate the cavitation effect of microbubbles can increase blood–tumor barrier permeability and vascular perfusion, significantly increasing the diffusion of agents and sensitizing chemoresistance.

**Figure 2 pharmaceutics-14-01642-f002:**
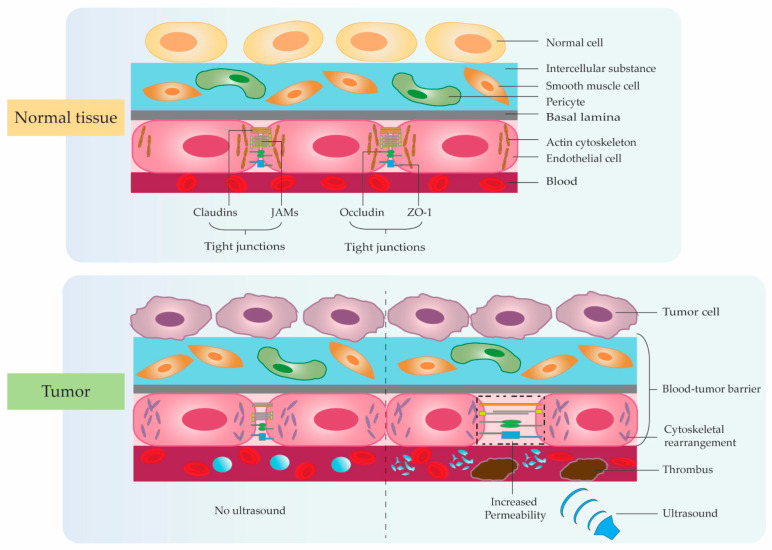
Ultrasound-mediated microbubble cavitation enhancing tumor–blood barrier permeability. The blood–tumor barrier is composed of vascular endothelial cells, basement membrane, and tumor interstitial cells. Biological barrier regulator proteins include Occludin, Claudin, JAM, ZO, and so on. Ultrasound-mediated microbubble cavitation can significantly increase local tissue tight junction protein permeability and increase drug diffusion, meanwhile, can form vascular microcirculation thrombus by further increasing ultrasound frequency to induce tumor ischemic necrosis.

**Table 1 pharmaceutics-14-01642-t001:** Ultrasound-mediated microbubble adjuvant drug therapy in vitro.

Cavitation Mechanism	Authors	Cell Type	Component	Intervention	Outcomes after Cavitation Effect
Enhanced permeability	Tinkov et al. [105]	Renal carcinoma cell	Doxorubicin	Group 1: DOX Group 2: DOX + MBs	Approximately two-fold enhanced anti-proliferative effect in DOX-loaded MBs. DOX-loaded MBs with high affinity to the nucleus.
Enhanced permeability Promoted drug diffusion	Li F. et al. [106]	Renal carcinoma cell	rAAV	Group 1: rAAV Group 2: rAAV + MBs Group 3: rAAV + US Group 4: rAAV + UTMD	US-mediated MBs inhibit tumor cell proliferation and induce apoptosis. US-mediated MBs promote viral transfection approximately two-fold.
Enhanced permeability Promoted drug diffusion	Haag P. et al. [107]	Prostate tumor cell	ODNs	Group 1: MBs Group 2: ODNs Group 3: ODNs + MBs Group 4: ODNs + US Group 5: ODNs + MBs + US	Best US frequency: 1.75 MHz; best MI: 1.9. US-mediated MBs inhibit AR protein levels by 36.23%. US-mediated MBs promote viral transfection approximately 40-fold.
Promoted drug uptake	Yan F. et al. [108]	Breast cancer cell	Paclitaxel and LyP-1 Peptide	Group 1: MBs Group 2: PTX-loaded MBs Group 3: Targeted PTX-loaded MBs	Targeted ultrasonic MBs encapsulation rate: 63%. US for 2 min increased cell uptake approximately seven-fold.
Enhanced permeability Promoted drug diffusion	Cochran M.C. et al. [109]	Breast cancer cell	Doxorubicin and paclitaxel	Group 1: MBs Group 2: MBs + US Group 3: Drug-loaded MBs Group 4: Drug-loaded MBs + US	The encapsulation efficiency of PTX and DOX: 72%, 20.5%. The payload of PTX-loaded MBs is 20 times DOX. The anti-tumor effect increased by 80.1%.
Promoted drug diffusion	Wang D.S. et al. [110]	Vascular endothelial tumor cell	DNA	Group 1: Cationic MBs + US Group 2: Neutral MBs + US Group 3: US Group 4: Cationic MBs	Cationic MBs protect plasmid DNA from degradation. Cationic MBs promote gene transfection approximately two-fold.
Enhanced permeability	Ren S.T. et al. [111]	Colon adenocarcinoma cell	Docetaxel	Group 1: DOC Group 2: DOC + US Group 3: MBs + US Group 4: DOC + MBs + US	The maximum encapsulation rate: 54.9%. The anti-tumor effect increased approximately two-fold.
Enhanced permeability Promoted drug diffusion	Escoffre J.M. et al. [112]	Glioblastoma cell	Doxorubicin	Group 1: MBs + US Group 2: DOX + MBs Group 3: DOX + MBs + US	US-mediated MBs significantly increased drug uptake. Tumor cell death with US-mediated MBs was enhanced approximately three-fold.

Abbreviations: MBs, microbubbles; DOX, doxorubicin; US, ultrasound; PTX, paclitaxel; DOC, docetaxel; rAAV, recombinant adeno-associated virus; UTMD, ultrasound-targeted microbubble destruction; ODNs, oligodeoxynucleotides; MI, mechanical index; AR, androgen receptor.

**Table 2 pharmaceutics-14-01642-t002:** Ultrasound-mediated microbubble adjuvant drug therapy in vivo.

Cavitation Mechanism	Authors	Cell Type	Component	Intervention	Outcomes after Cavitation Effect
Enhanced permeability	Wang G. et al. [113]	Hepatic cancer	Evans Blue	Group 1: EB Group 2: EB + MBs Group 3: EB + US Group 4: EB + MBs + US	US-mediated MBs cavitation can increase tumor vascular permeability. The cavitation effect promotes drug release approximately three-fold.
Enhanced permeability	Tang Q. et al. [114]	Hepatic cancer	HSV-TK/GCV	Group 1: pEGFP-KDR-TK + pEGFP-C1-AFP-TK Group 2: pEGFP-KDR-TK + pEGFP-C1-AFP-TK + US Group 3: pEGFP-KDR-TK + pEGFP-C1-AFP-TK + MBs + US	US-mediated MBs can increase killing effect of HSV-TK/GCV and CD/5-FC systems on vascular and hepatoma cells. US-mediated MBs can increase tumor vascular permeability and gene transfection efficiency.
Enhanced permeability Induced tumor necrosis	Li P. et al. [115]	Subcutaneous VX2 cancer	Evans Blue	Group 1: EB Group 2: EB + MBs Group 3: EB + US Group 4: EB + MBs + US	US-mediated MBs can induce tumor microvasculature disruption resulting in hemorrhage, edema, and thrombosis to cause necrosis.
Enhanced permeability	Cool S.K. et al. [100]	No tumor	ICG-liposomes	Group 1: Drug-MBs + US Group 2: Drug + MBs + US Group 3: MBs + US Group 4: Drug + US	MBs can increase ICG-liposomes loaded approximately three-fold. US-mediated MBs increase drug release two times more. US-mediated MBs can cause skin lesions due during microbubble collapse.
Enhanced permeability Promoted drug diffusion Enhanced perfusion	Lin C.Y. et al. [116]	Colon cancer	DOX	Group 1: DOX Group 2: DOX + MBs Group 3: DOX + US Group 4: DOX + MBs + US	US-mediated MBs cavitation can increase tissue permeability and destroy tumor vessels. US-mediated MBs can increase tumor drug uptake and inhibit growth. US intervention time should be less than 2 min.
Enhanced permeability Promoted drug diffusion	Fokong S. et al. [117]	Colon cancer	Rhodamine-B Coumarin-6	Group 1: MBs-Rhodamine-B Group 2: MBs-Coumarin-6 Group 3: MBs-Rhodamine-B + US Group 4: MBs-Coumarin-6 + US	The polymer-based MBs are highly suitable for image-guided, targeted, and triggered drug delivery to tumors and blood vessels.
Enhanced permeability Induced tumor necrosis	Huang P. et al. [65]	Colon cancer	No drug	Group 1: MBs Group 2: US Group 3: MBs + US	US-mediated MBs inhibit tumor growth and metastasis. US-mediated MBs destroy tumor cell nucleus and microvascular. US-mediated MBs decreases the expression of CD31.

Abbreviations: EB, Evans Blue; MBs, microbubbles; DOX, doxorubicin; US, ultrasound; HSV-TK/GCV, Herpes simplex virus-thymidine kinase/ganciclovir; KDR, kinase insert domain receptor; 5-FC, 5-fluorocytosine; TK, thymidine kinase; pEGFP, enhanced green fluorescent protein plasmid; ICG, indocyanine green; PBCA, poly (butyl cyanoacrylate).

## Data Availability

Not applicable.

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
