# Peer review of "Ultrasonic Microbubble Cavitation Enhanced Tissue Permeability and Drug Diffusion in Solid Tumor Therapy"

_pharmaceutics, 2022, doi:10.3390/pharmaceutics14081642_

Round 1

Reviewer 1 Report

Dear Authors,

your review is very interesting. It provides the reader with clear insights into the scientific topic of great interest in medical oncology. I think that your paper will be accept in present form.

Best regards

Author Response

Thank you for your reviewer’s comments concerning our study by He and colleagues. We are very grateful for your valuable comments on the review.

Reviewer 2 Report

The manuscript describes the application of microbubbles and ultrasound to improve the anti-cancer therapy. The topic is new and promising. The article gives the deep insight in this question using 105 references in corresponding journals. Step-by-step the authors describes the advantiges and questions of microbubble cavitation technique for increase of environment exchange in cancer tissues - first of all, in vessels. The article is very well written and easy to read and understand. The article is well illustrated by corresponding pictures. The review is consize, the conclusions are appropriate. As for English, I am not native speaker, for me English is very good. I did not find any serous mistakes to mention. I think the article can be published in present form.

Author Response

I am very honored to receive your reply, thank you for reviewing my manuscript during your busy schedule, thank you for your comments and recognition of my review. 

Reviewer 3 Report

General comment:

The paper of He and colleagues reviews the application of ultrasonic microbubble cavitation to counteract chemoresistance in solid tumors. Although the authors refer that this review paper discusses the progress and the mechanisms of action of this approach, these topics should be further exploited and discussed before publication.

Major issues:

1-       This review needs extensive English proofreading. 

2-       The manuscript should be reorganized to facilitate its reading and understanding. The sentences are very long and not easy to follow. Please break down the long sentences.

3-       The process of cavitation is described in lines 139 and 150. This concept should be clearly described at the beginning of section 3. Authors should briefly describe the process of ultrasonic microbubble cavitation in section 3.1., and then present the data regarding its application in tumor therapy.

4-       In line 156, the authors refer that ultrasound therapy can affect the active transport of drugs. However, they did not explain how the transport of drugs is affected. It is enhanced/inhibited? Affects the functionality of efflux or influx transporters expressed at the blood-tumor barrier? Considering that the expression of efflux transporters such as ATP-binding cassette (ABC) transporters is a crucial chemoresistance mechanism in cancer, it would be important to exploit this point.

5-   In lines 177-178, barrier structures are enumerated. Please correct and substitute “testicular capillary endothelial cells” with “blood-testis barrier” or “Sertoli cell seminiferous epithelium barrier” since is established by Sertoli cells (doi: 10.1124/pr.110.002790). The Blood-brain barrier should be also referenced since is one of the most important biological barriers and hinders the access of many therapeutic molecules to the central nervous system.

6-       Authors refer that ultrasonic microbubble cavitation presents some side effects such as hemorrhage, thrombus, and tissue necrosis, among others (line 210). Therefore, what are the reasons that justify its application? Could these side effects be mitigated through further development of the technique and how?

7-   Tables 1 and 2 present interesting data concerning the application of ultrasound-mediated microbubble therapy in combination with other therapeutical approaches in vitro and in vivo. Despite this, the information could be better organized to facilitate the reading. The rows “Intervention” and “Outcomes” are not aligned. Maybe remove the "Intervention" column or align the different outcomes with the corresponding "Intervention Group".

8-       Authors need to give more emphasis on the chemoresistance mechanisms affected by the ultrasonic microbubble cavitation in the manuscript, including those referred to in tables 1 and 2 (barrier permeability, drug diffusion, drug uptake).

 Minor issues:

1- Please substitute “scholars” by “researchers” in the manuscript.

2- Tumor microenvironment appears either under the abbreviature “TME” or undefined. Please uniformize.

3-    Please substitute “biofilm barriers” by “biological barriers” in the manuscript.

4-  Figure 2 does not show the decreased expression of TJs after the ultrasound. Please correct.

Author Response

Reviewer #3:

Thank you for your reviewer’s comments concerning our study by He and colleagues. We are very grateful for your valuable comments on the review. The following is my response to the question.

Question 1: This review needs extensive English proofreading.

Answer 1: We thank the reviewer for the good comment. We have double-checked the English to avoid grammar and spelling errors. Besides, we have used some online websites to polish the manuscript. And we strive to make the manuscript expression more accurate and concise.

Question 2: The manuscript should be reorganized to facilitate its reading and understanding. The sentences are very long and not easy to follow. Please break down the long sentences.

Answer 2: It’s really true as reviewer suggested that manuscripts sentences should not be too long. We have read the instructions for authors carefully again, especially for the review article section. We believe that the length of text sentences in this manuscript indeed meet the requirements of Pharmaceutics. However, we still carefully revised the entire manuscript repeatedly. We have shortened manuscript sentences content and used some online websites to polish the manuscript. We strive to make the manuscript expression more accurate and concise.

Question 3: The process of cavitation is described in lines 139 and 150. This concept should be clearly described at the beginning of section 3. Authors should briefly describe the process of ultrasonic microbubble cavitation in section 3.1., and then present the data regarding its application in tumor therapy.

Answer 3: Thanks for the Reviewer’s suggestion to improve better reasonable chapter content. We have added a brief description of ultrasonic microbubble cavitation and therapy application in section 3.1. in the revised manuscript (line 118-124). Please review them and give your valuable comments.

Question 4: In line 156, the authors refer that ultrasound therapy can affect the active transport of drugs. However, they did not explain how the transport of drugs is affected. It is enhanced/inhibited? Affects the functionality of efflux or influx transporters expressed at the blood-tumor barrier? Considering that the expression of efflux transporters such as ATP-binding cassette (ABC) transporters is a crucial chemoresistance mechanism in cancer, it would be important to exploit this point.

Answer 4: Thank you for your careful inspection and pointing out the negligence on active transport effect. We have added a detailed description of active transport pathway mechanism on cavitation effect in the revised manuscript (line 167-178). Active transport pathway is associated with molecular reaction including membrane resealing or gap restoration in different spatial and temporal scales, resulting in increased intracellular Ca2+ concentration and cytoskeletal rearrangement to induce clathrin-mediated endocytosis pathway to promote drug diffusion into cells. Regarding the reviewer’s comments that whether the ATP-binding cassette (ABC) transporters with chemoresistance mechanism could affect the acoustic cavitation effect, I have tried to consult the relevant literature many times but, unfortunately, I could not found any current research evidence to support the involvement of ATP-binding cassette (ABC) transporters pathway in the regulation of cavitation effects. This evidence from reviewer’s comments in a certain degree remind me to pay more attention to the regulation mechanism in the future research.

Question 5: In lines 177-178, barrier structures are enumerated. Please correct and substitute “testicular capillary endothelial cells” with “blood-testis barrier” or “Sertoli cell seminiferous epithelium barrier” since is established by Sertoli cells (doi: 10.1124/pr.110.002790). The Blood-brain barrier should be also referenced since is one of the most important biological barriers and hinders the access of many therapeutic molecules to the central nervous system.

Answer 5: We quite agree with this suggestion. We have revised the manuscript based on the content of the full chapter. After double-check and thought, I consider it is indeed more appropriate to replace "testicular capillary endothelial cells" with "blood-testis barrier". Thanks for comments on article you recommended because of beneficial improvement a lot after reading it. I consider this revise will be significant for the scientific rigor of this paper. Also, I have added description cited of the "blood-brain barrier" in this chapter.

Question 6: Authors refer that ultrasonic microbubble cavitation presents some side effects such as hemorrhage, thrombus, and tissue necrosis, among others (line 210). Therefore, what are the reasons that justify its application? Could these side effects be mitigated through further development of the technique and how?

Answer 6: It’s really true as reviewer suggested that we should pay more attention to side effects of intervention strategy. We have double-checked this section and added a detailed description of side effect by utilizing acoustic cavitation effect (line 242-250). Thankfully, most of the serious side effects of ultrasound-assisted therapy including necrosis and hemorrhage, are concentrated in a relatively high intensity focused ultrasound which mainly exerts acoustic thermal effect rather than cavitation effect. Conversely, when we controlled reasonably the ultrasonic to maintain lower intensity level and shorter intervention time, the cavitation effect could induce variable degrees of cavitation effect without obvious cell death and thermal damage. Overall, low-intensity ultrasound is relatively safe non-invasive intervention strategy. It is worthwhile to expect that flexible regulation of ultrasonic intervention parameters and better protocol design can further improve cavitation effect efficacy and potential risk aversion.

Question 7: Tables 1 and 2 present interesting data concerning the application of ultrasound-mediated microbubble therapy in combination with other therapeutical approaches in vitro and in vivo. Despite this, the information could be better organized to facilitate the reading. The rows “Intervention” and “Outcomes” are not aligned. Maybe remove the "Intervention" column or align the different outcomes with the corresponding "Intervention Group".

Answer 7: Thank you for taking the time to review my manuscript and provide valuable comments. Regarding your question 7, in fact, our consideration on this issue is as follows. From our edited Tables 1 and 2, the column "Intervention" indicates the experimental groupings in the literature, and the column "Outcomes" indicates the results and significance of this study in the application of ultrasonic cavitation. In order to better acknowledge and present the value of the ultrasonic cavitation effect in this study, we corresponded "Intervention" and "Outcomes" in the same dimension. Combined with the valuable comments you mentioned, your suggestion in my consideration that we should present the conclusions and experimental results of all groups in each article in the "Outcomes" column. However, considering the large amount of information and research conclusions contained in each article, we may not have much space in this table to display all grouping information for each article. But I still think your comments are correct and reasonable, so we decided to alter the "Outcomes" in the revised manuscript to "Outcomes after cavitation effect" based on your comments for easier understanding to readers.

Question 8: Authors need to give more emphasis on the chemoresistance mechanisms affected by the ultrasonic microbubble cavitation in the manuscript, including those referred to in tables 1 and 2 (barrier permeability, drug diffusion, drug uptake).

Answer 8: Thanks for the Reviewer’s reminder. We have added a detailed description of ultrasonic microbubble cavitation effect in the revised manuscript (line 214-223). Overall, ultrasound can utilize the local microbubble cavitation to enhance the EPR effect for non-invasive targeted therapy of diseases without affecting the surrounding soft tissues. The cavitation effect can achieve the passive agents diffusion in tissue through ameliorating the permeability of tissue and vascular barrier by sonoporation and regulate the intercellular tight junction. Similarly, it can also enhance intracellular uptake via stimulating the clathrin-mediated endocytosis pathway induced by ion channels. On the one hand, the cavitation effect can also enhance the drugs delivery efficiency by inducing the releasing vasodilator to a certain extent to increase local tissue blood perfusion. The feasibility and potential of this approach might contribute to achieve better targeted delivery in the prospective fast-revolutionizing disease area.

Question Minor issues 1-4:

1- Please substitute “scholars” by “researchers” in the manuscript.

2- Tumor microenvironment appears either under the abbreviature “TME” or undefined. Please uniformize.

3- Please substitute “biofilm barriers” by “biological barriers” in the manuscript.

4- Figure 2 does not show the decreased expression of TJs after the ultrasound. Please correct.

Answer minor issues 1-4: We thank the reviewer for the good comment. We have double-checked the manuscript content based on your suggestions and recorrect the minor issues including “researcher”, “biological barriers”, “expression of TJs after the ultrasound in Figure 2” and “TME” to make sure the manuscript expression more accurate and concise.

Round 2

Reviewer 3 Report

The authors addressed most of the reviewers considerations and suggestions. 

Still, moderate/minor English changes should be performed.